# Developing and validating an individualized breast cancer risk prediction model for women attending breast cancer screening

Javier Louro[1,2,3,4], Marta Román[1,2,3]*, Margarita Posso[1,2,3], Ivonne Vázquez[5], Francina Saladié[6], Ana Rodriguez-Arana[7], M. Jesús Quintana[8,9], Laia Domingo[1,2,3], Marisa Baré[2,10], Rafael Marcos-Gragera[9,11], María Vernet-Tomas[12], Maria Sala[1,2,3], Xavier Castells[1,2,3], on behalf of the BELE and IRIS Study Groups[¶]

1 IMIM (Hospital del Mar Medical Research Institute), Barcelona, Spain, 2 Research Network on Health Services in Chronic Diseases (REDISSEC), Barcelona, Spain, 3 Servei d'Epidemiologia i Avaluació, Hospital del Mar, Barcelona, Spain, 4 European Higher Education Area (EHEA) Doctoral Programme in Methodology of Biomedical Research and Public Health in Department of Pediatrics, Obstetrics and Gynecology, Preventive Medicine and Public Health, Universitat Autónoma de Barcelona (UAB), Bellaterra, Barcelona, Spain, 5 Servei de Patologia, Hospital del Mar, Barcelona, Spain, 6 Cancer Epidemiology and Prevention Service, Hospital Universitari Sant Joan de Reus, Institut d'Investigació Sanitària Pere Virgili, Universitat Rovira i Virgili, Reus, Spain, 7 Servei de Diagnòstic per la imatge, Hospital del Mar, Barcelona, Spain, 8 Department of Clinical Epidemiology and Public Health, University Hospital de la Santa Creu i Sant Pau, IIB Sant Pau, Barcelona, Barcelona, Spain, 9 CIBER of Epidemiology and Public Health (CIBERESP), Barcelona, Spain, 10 Clinical Epidemiology and Cancer Screening, Parc Taulí University Hospital, Sabadell, Spain, 11 Department of Health, Epidemiology Unit and Girona Cancer Registry, Oncology Coordination Plan, Autonomous Government of Catalonia, Catalan Institute of Oncology, Girona, Spain, 12 Servei d'Obstetricia i Ginecologia, Hospital del Mar, Barcelona, Spain

¶ Complete membership of the author group can be found in the Acknowledgments
* mroman@parcdesalutmar.cat

**Data Availability Statement:** Wwe have uploaded the database to the Harvard Dataverse online

## Abstract

### Background

Several studies have proposed personalized strategies based on women's individual breast cancer risk to improve the effectiveness of breast cancer screening. We designed and internally validated an individualized risk prediction model for women eligible for mammography screening.

### Methods

Retrospective cohort study of 121,969 women aged 50 to 69 years, screened at the long-standing population-based screening program in Spain between 1995 and 2015 and followed up until 2017. We used partly conditional Cox proportional hazards regression to estimate the adjusted hazard ratios (aHR) and individual risks for age, family history of breast cancer, previous benign breast disease, and previous mammographic features. We internally validated our model with the expected-to-observed ratio and the area under the receiver operating characteristic curve.

### Results

During a mean follow-up of 7.5 years, 2,058 women were diagnosed with breast cancer. All three risk factors were strongly associated with breast cancer risk, with the highest risk

repository. The data is accessible with DOI: https://doi.org/10.7910/DVN/3T7HCH.

**Funding:** This study was supported by grants from Instituto de Salud Carlos III FEDER [PI15/00098 and PI17/00047]; the Research Network on Health Services in Chronic Diseases [RD12/0001/0015]; and the Spanish Society of Epidemiology (SEE) [XV Alicia Llacer grant for the best research by a young researcher].

**Competing interests:** The authors have declared that no competing interests exist.

**Abbreviations:** aHR, Adjusted hazard ratio; AUC, Area under the receiving operating characteristic curve; BI-RADS, Breast Imaging Reporting and Data System; BBD, Benign breast disease; DCIS, Ductal carcinoma in situ; E/O, Expected to observed; SNPs, Single Nucleotide Polymorphisms; 95%CI, 95% Confidence intervals.

being found among women with family history of breast cancer (aHR: 1.67), a proliferative benign breast disease (aHR: 3.02) and previous calcifications (aHR: 2.52). The model was well calibrated overall (expected-to-observed ratio ranging from 0.99 at 2 years to 1.02 at 20 years) but slightly overestimated the risk in women with proliferative benign breast disease. The area under the receiver operating characteristic curve ranged from 58.7% to 64.7%, depending of the time horizon selected.

## Conclusions

We developed a risk prediction model to estimate the short- and long-term risk of breast cancer in women eligible for mammography screening using information routinely reported at screening participation. The model could help to guiding individualized screening strategies aimed at improving the risk-benefit balance of mammography screening programs.

## Introduction

There is ongoing debate on the benefits and harms of breast cancer screening [1–3]. To improve this balance, current evidence supports personalized screening [4,5]. Modeling studies have shown that modifying the screening interval, screening modality, or age range of the target population based on women's individual risk yielded greater benefit than conventional standard strategies [5–7]. Several risk models have been designed to estimate women's individual breast cancer risk based on their personal characteristics [8–15]. However, most of these models have not been specifically developed to estimate the risk of women targeted for breast cancer screening in order to offer them personalized strategies.

A recent consensus statement of the European Conference on Personalized Early Detection and Prevention of Breast Cancer (ENVISION) [16] stated the need to develop breast cancer risk prediction models based on data from large screening cohorts and including risk factors easily obtainable at screening participation, such previous mammographic features and prior benign breast disease.

To date, only one model has specifically aimed to predict women's individual risk looking to personalize breast cancer screening strategies [17]. Although highly valuable, the model was based on short-term risk estimates and did not account for relevant characteristics of prospective studies such as internal time-dependent covariates. This model only estimates the two-year risk, which could lead to bias as one of the aims proposed in breast cancer screening personalization is to see which women are at a lower risk in order to extend their screening period to three or four years. Therefore, if new breast cancer risk models are developed with the aim of analyzing the possibilities offered by personalized screening strategies, it would be interesting to estimate the biennial risk of each woman, in other words, to obtain estimators not only at 2 years, but also every two years (2, 4, 6, 8. . . up to 20 years, which is the total time a woman is screened). This will help to better understand the different possibilities of screening strategies and will allow to observe the differences in the validation of the model estimators for the different time horizons. There is therefore a need for breast cancer risk prediction models, with risk estimates in the short- and long-term, and based on data from large screening cohorts. These new risk models should include a limited and feasible number of variables for the proposed objective, for example, detailed information on the type of previous benign

breast disease or previous mammographic characteristics, which existing risk models tend not to use.

We aimed to design and validate an individualized risk prediction model to estimate the biennial risk of breast cancer in women eligible for mammography screening by using data from the long-standing population-based screening program in Spain.

## Materials and methods

### Setting and study population

Breast cancer screening in Spain started in 1990 in a single setting and expanded until it became nationwide in 2006. This program follows the recommendations of the European Guidelines for Quality Assurance in Breast Cancer Screening and Diagnosis [18]. Women aged 50 to 69 years are invited to biennial screening mammography by written letter. Screening mammograms are interpreted according to the Breast Imaging Reporting and Data System (BI-RADS) scale by trained breast radiologists [19]. Women with an abnormal mammographic feature are recalled for further assessments to confirm or rule out malignancy. Women without a breast cancer diagnosis are invited again for routine screening at 2 years. Overall, breast cancer screening in Spain has a recall rate of 43.0, a detection rate of 4.0, and an interval cancer rate of 1.1 per 1,000 mammographic examinations [20]. The positive predictive value is 9.8% for recalls and 38.9% for recalls involving invasive procedures. Overall, 16.8% of all screen-detected cancers are ductal carcinoma in situ (DCIS). More details of breast cancer screening in Spain are described elsewhere [21].

We analyzed data from two centers forming part of the Spanish breast cancer screening program in the Metropolitan Area of Barcelona. These centers routinely gather information on family history of breast cancer, previous benign breast disease (BBD), and previous mammographic features. The centers collect information on screening mammography examinations, recalls, further assessments, and diagnostic results in their defined catchment areas. The cohort included all 123,251 women screened at least once between 1995 and 2015 and followed-up until December 2017. We excluded 758 women diagnosed with breast cancer at the first screen, 210 women with missing information on family history, 213 women with missing information on previous BBD, and 101 women with missing information for both family history and previous BBD. The study population for the analysis consisted of 121,969 women who underwent 437,540 screening mammograms during the study period.

### Definition of study variables

Information on family history and history of prior breast biopsies was self-reported and collected from face-to-face interviews conducted by trained professionals at the time of mammography screening. This information was consistently collected over the 20 years study period. A family history of breast cancer was defined as having at least one first-degree relative with a history of breast cancer.

Breast biopsy results were classified by a community pathologist at each center using SNOMED codes [22]. Pathological diagnoses were grouped following the benign breast disease classification proposed by Dupont and Page [23–25] into non-proliferative and proliferative disease. Proliferative lesions with and without atypia were combined into a single category due to the small number of subsequent breast cancer cases among those with a proliferative lesion with atypia. If women reported having had a biopsy before the start of the screening but no pathology results were available, the biopsy was classified as having a prior biopsy, unknown diagnosis.

A community radiologist routinely reported on mammographic features found at mammography screening interpretation. We classified as mammographic features any mass, calcification, asymmetry or architectural distortion reported by radiologists at mammographic interpretation. Findings were assigned to the category of multiple mammographic features if more than one of the previous mammographic features had been reported simultaneously at screening interpretation.

We included both invasive breast cancers and DCIS for the analysis.

## Model design

We built the risk prediction model using a random sample of 60% of the study population (estimation subcohort). The remaining 40% was used for an internal validation (validation subcohort).

We estimated the age-adjusted hazard ratios (aHR) and the 95% confidence intervals (95%CI) for the breast cancer incidence for each category of family history, previous BBD, and previous mammographic features with the estimation subcohort. Age was included in the model as a continuous variable. We used partly conditional Cox proportional hazards regression, an extension of the standard Cox model, to incorporate changes in these risk factors over time. Robust standard errors were used to estimate 95% confidence intervals using the Huber sandwich estimator [26]. If a woman has had a diagnosis of cancer, she will contribute women-years at risk from the date of her first mammogram to the diagnosis of cancer. Since we can identify all interval cancers, a woman who has not had a diagnosis of cancer at the end of her follow-up will contribute women-years at risk from the first mammogram to the last mammogram plus 2 years of follow-up.

We tested whether family history, previous BBD, and previous mammographic features interacted among themselves or with age. The interaction terms were not significant and were therefore not included in the model. The proportional hazards assumption was assessed by plotting the log-minus-log of the survivor function against log time for each predictor variable. The proportional hazards assumption appeared to be reasonable for all predictors.

## Model validation

We calculated the absolute breast cancer risk estimates for each 2-year interval over the 20-year lifespan covered by screening (ages 50 to 69 years) for each individual in the validation subcohort. As proposed by Zheng and Heagerty, we used a general hazard function to predict the absolute risk of breast cancer diagnosis based on length of follow-up, prediction time, and women's risk profile [27].

We conducted an internal validation of the model to evaluate its predictive performance by assessing its calibration and discrimination. To assess calibration, we calculated the ratio between the expected breast cancer rate in the validation subcohort versus the observed rate in the estimation subcohort. To account for censoring, the observed rate was estimated using the Kaplan-Meier estimator. The expected breast cancer rate was calculated as the average of the risk estimates in the validation subcohort. The expected breast cancer rate in a specific risk group was calculated as the average of the risk estimates for each woman in that risk group of the validation subcohort. The expected-to-observed (E/O) ratio assessed whether the number of women predicted to develop breast cancer from the model matched the actual number of breast cancers diagnosed in the validation subcohort. An E/O ratio of 1.0 indicates perfect calibration. We calculated the E/O ratio 95% confidence intervals (95% CI) using the formula of the standardized mortality ratio proposed by Breslow and Day [28]. The discriminatory accuracy of our model was assessed by estimating the area under the receiving operating characteristic curve (AUC) for each 2-year interval based on the predicted risks for each woman and

whether she developed breast cancer during the time interval or not [29]. The predicted risks were calculated using the model coefficient estimates at the baseline mammogram for those women in the validation cohort who have been followed for a time greater than or equal to the time horizon being estimated. The AUC measured the ability of the model to discriminate between women who will develop breast cancer from those who will not. We calculated the 95% CI using the approach proposed by Hanley and McNeil [30].

Statistical tests were two-sided and all p-values <0.05 were considered statistically significant. All analyses were performed using the statistical software R version 3.4.3 (Development Core Team, 2014).

The study was approved by the Clinical Research Ethics Committee of Hospital del Mar Medical Research Institute (2015/6189/I). The review boards of the institutions providing data granted approval for data analyses. This is an entirely registry-based study that used anonymized retrospective data and hence there was no requirement for written informed consent.

The authors declare that they have no conflicts of interest.

## Results

During a mean follow-up of 7.52 years, breast cancer was diagnosed in 2,058 out of the 121,969 women in the study population. The mean follow-up was shorter in women with a breast cancer diagnosis than in those without (5.8 years vs 7.6 years, p-value < 0.05). Women with breast cancer were more likely to have a family history of breast cancer (18.32% vs 13.86%), biopsies with unknown diagnosis (23.76% vs 21.72%), non-proliferative and proliferative BBD (5.59% vs 3.20%, and 1.60% vs 0.45%, respectively), masses (20.51% vs 18.12%), and calcifications (6.85% vs 2.71%) (Table 1).

Breast cancer was strongly associated with previous benign breast disease, with the highest risk being found among women with a proliferative BBD (aHR, 3.02; 95% CI: 1.75, 5.21) compared with those without a BBD (Table 2). Family history was also associated with breast cancer (aHR, 1.67; 95% CI: 1.41, 1.98). Among women with previous mammographic features, the highest risks were found in calcifications (aHR, 2.52; 95% CI: 1.93, 3.29) and architectural distortions (aHR, 2.07; 95% CI: 1.27, 3.38).

Overall calibration of the model was accurate across all 2-year time horizons. The E/O ratio ranged from 0.99 at 2 years to 1.02 at 20 years and was never significantly different than 1 (Table 3). The AUC was lowest at the 4-year risk estimate (AUC, 58.7%; 95%CI: 55.9%-61.5%) and highest at the 18-year risk estimate (AUC, 64.7%; 95%CI: 62.5%-66.9%) and were significantly higher than 50% for all the time horizons.

Estimates for the 10-year time horizon showed that the model slightly overestimated breast cancer rates in women with masses (E/O ratio, 1.18; 95%CI: 1.02–1.37) and in women aged 55–59 years (E/O ratio, 1.15; 95%CI: (1.03–1.29) (Table 4). The model also underestimated breast cancer rates in women aged 50–54 years (E/O ratio, 0.83; 95%CI: 0.75–0.94). Because of the small number of breast cancer cases, calibration was overestimated among women with proliferative BBD (E/O ratio, 1.85; 95%CI: 1.00–3.40).

Distribution of the absolute cumulative risk estimates at 2-, 10- and 20-year time horizons are shown in Fig 1. The 10-year risk was between 1.5% and 2% in 60% of the women and was higher than 2% in 35%. The 20-year risk was lower than 3% in only 4% of the women, between 5% and 7% in 17% of the women, and was higher than 7% in approximately 9% of the women.

## Discussion

We used individual-level data from a large cohort of women regularly screened in Spain to design and validate a risk prediction model to estimate the biennial risk of breast cancer in

**Table 1. Baseline characteristics of the study population.**

| | No breast cancer (n = 119,911) | Breast cancer (n = 2,058) | p-value |
|---|---|---|---|
| **Mean follow- up** | 7.6 years | 5.8 years | <0.001 |
| **Age (years)** | | | |
| 50–54 | 63,507 (52.96%) | 1,149 (55.83%) | 0.010 |
| 55–59 | 25,738 (21.46%) | 542 (26.34%) | <0.001 |
| 60–64 | 22,796 (19.01%) | 325 (15.79%) | <0.001 |
| 65–69 | 7,870 (6.56%) | 42 (2.04%) | <0.001 |
| **Family history of breast cancer** | | | |
| No | 103,296 (86.14%) | 1,681 (81.68%) | <0.001 |
| Yes | 16,615 (13.86%) | 377 (18.32%) | <0.001 |
| **Benign breast disease** | | | |
| None | 89,500 (74.64%) | 1,421 (69.05%) | <0.001 |
| Prior biopsy, unknown diagnosis | 26,042 (21.72%) | 489 (23.76%) | 0.028 |
| Non-proliferative | 3,832 (3.20%) | 115 (5.59%) | <0.001 |
| Proliferative | 537 (0.45%) | 33 (1.60%) | <0.001 |
| **Mammographic features** | | | |
| None | 86,326 (71.99%) | 1,283 (62.34%) | <0.001 |
| Mass | 21,728 (18.12%) | 422 (20.51%) | <0.001 |
| Calcifications | 3,246 (2.71%) | 141 (6.85%) | <0.001 |
| Asymmetry | 3,371 (2.81%) | 56 (2.72%) | 0.858 |
| Architectural distortion | 1,249 (1.04%) | 29 (1.41%) | 0.129 |
| Multiple features | 3,991 (3.33%) | 127 (6.17%) | <0.001 |

Differences in mean of follow-up were tested by Mann–Whitney U test.

Differences in qualitative variables were tested by two-sided test of equality for column proportions (z-test). Tests adjusted for all pairwise comparisons within each tumor characteristic using the Bonferroni correction.

**Table 2. Partly conditional Cox proportional hazards model results showing the hazard ratios of the risk factors on breast cancer.**

| | Women-years | Breast cancer cases | aHR* (95%CI) |
|---|---|---|---|
| **Family history of breast cancer** | | | |
| No | 471,552 | 976 | Ref. |
| Yes | 79,471 | 227 | 1.67 (1.41–1.98) |
| **Benign breast disease** | | | |
| No | 408,883 | 832 | Ref. |
| Prior biopsy, unknown diagnosis | 118,010 | 286 | 1.36 (1.16–1.59) |
| Non-proliferative | 21,123 | 67 | 1.41 (1.02–1.94) |
| Proliferative | 3,007 | 18 | 3.02 (1.75–5.21) |
| **Mammographic features** | | | |
| No | 380,314 | 752 | Ref. |
| Mass | 110,597 | 239 | 1.32 (1.11–1.57) |
| Calcifications | 17,160 | 81 | 2.52 (1.93–3.29) |
| Asymmetry | 17,526 | 38 | 1.66 (1.16–2.39) |
| Architectural distortion | 6,287 | 20 | 2.07 (1.27–3.38) |
| Multiple features | 19,140 | 73 | 1.86 (1.43–2.43) |

aHR: Adjusted Hazard Ratio. 95%CI: 95% Confidence Interval.

*Model adjusted by age, family history, previous benign breast disease and previous mammographic features.

**Table 3. E/O ratio and area under the ROC curve of the model for each time horizon.**

|  | Observed events | E/O ratio (CI95%) | AUC |
|---|---|---|---|
| 2-year risk | 188 | 0.99 (0.86–1.14) | 63.0 (59.1–66.9) |
| 4-year risk | 455 | 1.01 (0.92–1.11) | 58.7 (55.9–61.5) |
| 6-year risk | 685 | 1.00 (0.92–1.07) | 59.5 (57.2–61.8) |
| 8-year risk | 853 | 1.02 (0.95–1.09) | 61.0 (58.9–63.0) |
| 10-year risk | 1,000 | 1.01 (0.95–1.08) | 60.9 (59.0–62.8) |
| 12-year risk | 1,092 | 1.03 (0.97–1.09) | 60.5 (58.6–62.4) |
| 14-year risk | 1,165 | 1.01 (0.96–1.07) | 62.4 (60.5–64.3) |
| 16-year risk | 1,195 | 1.00 (0.95–1.06) | 64.3 (62.4–66.3) |
| 18-year risk | 1,201 | 1.01 (0.96–1.07) | 64.7 (62.5–66.9) |
| 20-year risk | 1,203 | 1.02 (0.97–1.08) | 63.8 (61.3–66.3) |

E/O: Expected observed. 95%CI: 95% Confidence Interval.

women aged 50 to 69 years eligible for mammography screening. We tested a model that uses only variables easily obtainable at screening participation. The model showed very good calibration but only modest discrimination.

Our model calculates the risk of breast cancer for each 2-year time horizon during a woman's screening lifespan. Until now, the 5-year risk estimate has been the standard since the BCRAT model used a 5-year risk time horizon for decision making about chemoprevention. The BCRAT model was the basis for enrolment into the two major US prevention trials

**Table 4. Calibration of the 10-year estimates from the model in risk factor subgroups.**

|  | Observed events | E/O ratio (95%CI) |
|---|---|---|
| **Overall** | 1,000 | 1.01 (0.95–1.08) |
| **Family history** | | |
| No | 824 | 1.00 (0.93–1.07) |
| Yes | 176 | 1.10 (0.95–1.28) |
| **Benign breast disease** | | |
| No | 709 | 1.02 (0.95–1.10) |
| Prior biopsy, unknown diagnosis | 238 | 1.02 (0.90–1.16) |
| Non-proliferative | 43 | 1.17 (0.87–1.57) |
| Proliferative | 10 | 1.85 (1.00–3.40) |
| **Mammographic features** | | |
| No | 661 | 1.03 (0.96–1.11) |
| Mass | 175 | 1.18 (1.02–1.37) |
| Calcifications | 60 | 1.07 (0.83–1.38) |
| Asymmetry | 29 | 1.14 (0.79–1.63) |
| Architectural distortion | 15 | 1.05 (0.63–1.73) |
| Multiple features | 60 | 0.90 (0.70–1.16) |
| **Age (years)** | | |
| 50–54 | 296 | 0.83 (0.75–0.94) |
| 55–59 | 296 | 1.15 (1.03–1.29) |
| 60–64 | 278 | 0.96 (0.85–1.08) |
| 65–69 | 130 | 1.02 (0.86–1.21) |

E/O: Expected observed. 95%CI: 95% Confidence Interval.

**A**

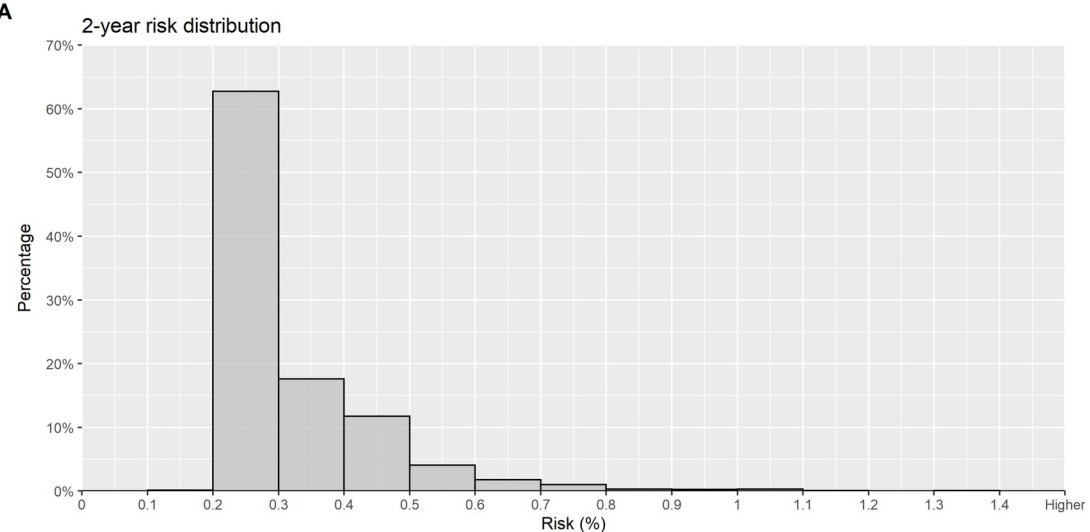

**B**

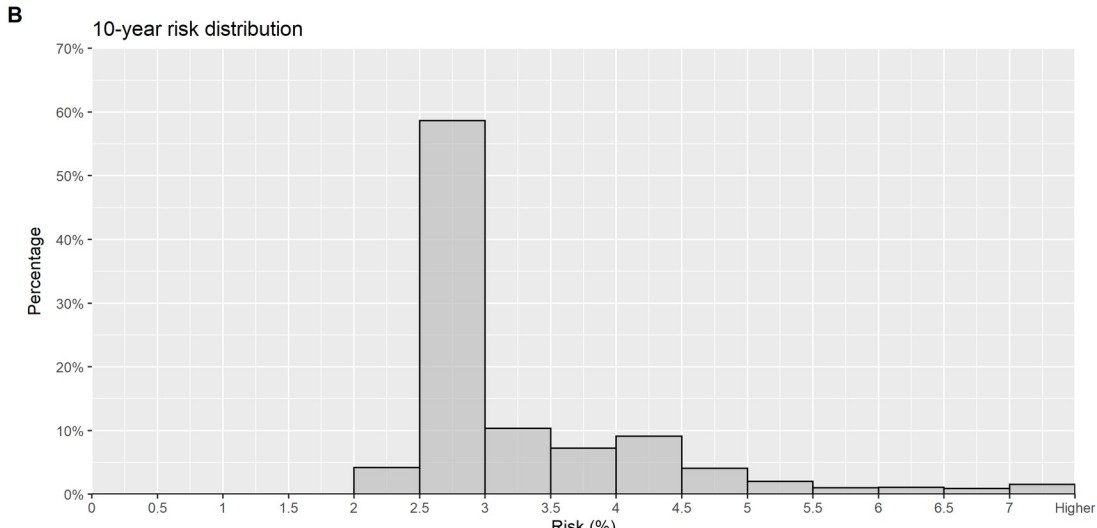

**C**

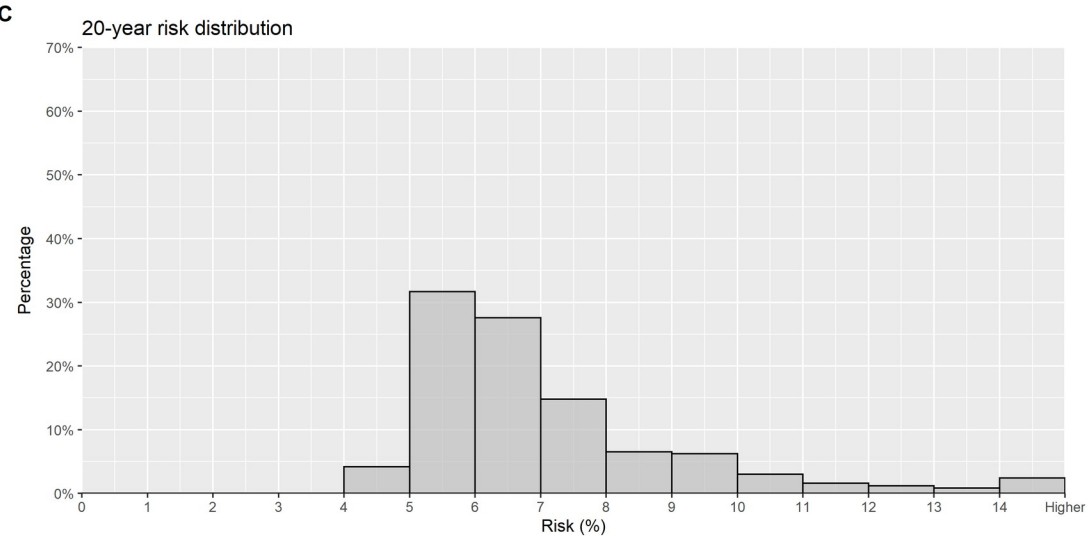

**Fig 1. Distribution of the absolute cumulative risk estimates.**

[31,32]. However, as stated in the statements of the last European Conference on Risk-Stratified Prevention and Early Detection of Breast Cancer, there is a need for risk models specifically designed for women eligible for breast cancer screening, based on data from large screening cohorts [16].

A previous model was designed to estimate the risk of breast cancer in women eligible for mammography screening [17]. The model used the Karma cohort from Sweden and included information on mammographic features. That study focused solely on estimating the short-term risk of breast cancer over the next mammographic examination. In addition, it used a case-control design to establish risk factors, which may bias the estimates of the short-term association with breast cancer risk. Our model adds to the breast cancer risk prediction models currently available and can be used to help guide personalized screening strategies by employing information easily obtained at screening participation. Additional useful information from our model is estimation of a woman's risk for breast cancer at 2-yearly intervals.

Our model was further developed by adding the effect of mammographic features, such as masses, calcifications, asymmetries, and architectural distortions. Previous studies have shown that mammographic features increase the subsequent risk of breast cancer [33]. In our model, the strongest influence on risk was conferred by calcifications. The biology behind calcifications is not well established. It has been suggested that mammary cells may acquire some mesenchymal characteristics, being able to contribute to the production of breast calcifications as a sign of carcinogenic transformation [34].

The role of BBD as a risk factor for breast cancer is well established [9,33,35]. However, its inclusion in breast cancer risk prediction models is rare, mainly because available information on BBD in large cohorts of women is uncommon. Only one previous risk model included different estimates for the different categories of the Dupont and Page BBD pathological classification [23–25]. The Breast Cancer Surveillance Consortium model was updated to include BBD, which led to only minimal improvement in discrimination [9]. This lack of significant improvement could be due to the absence of pathology results for most women who reported breast biopsies prior to their first screening round, as was also the case in our study. However, the addition of BBD to the model markedly increased the proportion of women identified as being at high risk for invasive breast cancer.

We assessed the internal validity of the model by means of its calibration and discriminatory accuracy. To perform internal validation we split our cohort in two sets, the estimation subcohort, to perform the analysis and development of the model and the validation subcohort, to perform the internal validation of the model. This technique known as split validation is common for this type of models [9] but cross validation or bootstrapping could also have been performed [36,37]. The model showed accurate calibration, neither overestimating nor underestimating the overall risk through the different years. In Table 4 we saw the calibration of the 10-year estimates from the model in risk factor subgroups. We also performed the E/O ratio estimates in risk factor subgroups for each one of the time horizons proposed. We only showed the 10-year estimates since showing all of them could be confusing. We showed the 10-year estimates since they have a good balance between the number of events observed (in the first time horizons some subcategories have a low number of observed events was lower) and the number of people observed (in the last time horizons we have some lost to follow–up, as the mean time of follow-up is 7.5 years). Nonetheless, the E/O ratio was overestimated for women with a proliferative BBD, due to the small number of cases among this subgroup.

The model showed modest discrimination with a maximum AUC of 64.7%. Discriminatory accuracy in breast cancer risk prediction models is usually low because a substantial proportion of cases are diagnosed in women with no known risk factors and the AUC of the different models vary between 60 and 70% [14]. This is clearly in contrast with prediction models for

other diseases, such as cardiovascular disease, which achieve good discrimination [38,39]. However, the model presented in this paper performed as well as other models that include many other risk factors that were not available in this study. As one of the reasons why the existing risk models have not been implemented for personalized screening is that it is difficult to collect all of the necessary risk factors in practice, a simpler model like the one we present could be useful. We tested other approaches to validate our model, such as the AUC estimation proposed by Li et al [40]. This estimation uses weights to calculate the contribution in the estimates of those women without a breast cancer diagnosis who were censored before reaching the time horizon. However, this approach produced no substantial differences in our validation.

A major strength of our model is that we used individual-level data from more than 120,000 women participating in a large, well-established, population-based screening program in Spain from 1995 to 2015, with a mean follow-up of more than 7.5 years and a maximum of 20 years. The program has a participation rate of 67% and a re-attendance rate of 91.2% [19].

This study also has some limitations. First, a major weakness is the lack of information on breast density, which was not systemically collected as part of screening data in the participating centers. Previous models estimating individual breast cancer risk have shown that the addition of breast density improved the discriminatory power of the models [9,17,41,42]. Dense breasts confer women a higher risk of breast cancer and are also associated with a higher risk of false-positive results, masking, and interval cancers [43]. In addition, we had no information on common genetic variants, which has been added to other breast cancer risk prediction models [44,45]. However, the discriminatory accuracy of the models was scarcely improved by the inclusion of information on single nucleotide polymorphisms (SNPs). This lack of both variables may be useful for some institutions where these risk factors are not available.

Second, the number of breast cancer cases among women with a proliferative BBD was small, which reduced our ability to accurately predict the expected number of cases across risk factor subgroups. Nevertheless, the overall calibration of the model across the time horizons assessed was highly accurate. Also, as a consequence of the small number of subsequent breast cancer cases among those women with a proliferative BBD with atypia, we merged proliferative BBD with and without atypia into a single category which might make the model less usable in practice.

Third, our model was based on a large set of representative data from the Breast Cancer Screening Program in Spain, which provides good generalizability. However, external validation of the results is needed to verify the predictive performance of our risk model.

Another limitation might be the reason for censoring. Over 52% of women in the cohort had their last mammogram in the last two years of the study follow-up and 17% of women had their last mammogram at ages 68 or 69 years. Most of the remaining 31% are women who did not participate in the 2014–2015 round or who have changed health areas and thus are not in our study population. The screening program does not have an exhaustive record of which women die and, therefore, we cannot differentiate them from non-participating women.

Finally, we were unable to analyze the association between the laterality of the BBD with the subsequent risk of breast cancer. In a previous analysis, we found that 40% of incident breast cancer cases in women with BBD were contralateral to the prior BBD, suggesting that a large proportion of benign lesions may be risk markers rather than precursors of subsequent cancer [46].

## Conclusions

We designed and internally validated a risk prediction model to estimate the short- and long-term risk of breast cancer in women eligible for mammography screening based on their age,

family history, previous benign breast disease, and previous mammographic features. The model showed good calibration and modest discriminatory power, and could be improved by adding further variables such as breast density and polygenic risk scores. The model can be used biennially to predict a woman's breast cancer risk during her screening lifespan (age 50 to 69 years) using information easily obtained at screening participation. Risk prediction models specifically designed for women eligible for breast cancer screening are key to guide individualized screening strategies aiming to improve the risk-benefit balance of mammography screening programs.

## Acknowledgments

The authors acknowledge the dedication and support of the Benign Lesion (BELE) Study Group leaded by Xavier Castells (xcastells@parcdesalutmar.cat) and listed here in alphabetical order and grouped by institution: (a) IMIM (Hospital Del Mar Medical Research Institute), Barcelona, Spain: Andrea Burón, Xavier Castells, Merce Comas, Jose Maria Corominas, Javier Louro, Ana Rodríguez-Arana, Marta Román, Maria Sala, Sonia Servitja, Mar Vernet-Tomas; (b) Corporació Sanitària Parc Taulí, Sabadell, Spain: Marisa Baré, Nuria Tora; (c) Catalan Institute of Oncology, Barcelona, Spain: Llucia Benito, Carmen Vidal (d) Hospital Santa Caterina, Girona, Spain: Joana Ferrer; (e) Catalan Institute of Oncology, Girona, Spain: Rafael Marcos-Gragera; (f) Hospital de la Santa Creu i Sant Pau, Barcelona, Spain: Judit Solà-Roca, María Jesús Quintana; (g) General Directorate of Public Health, Government of Cantabria, Spain: Mar Sánchez; (h) Principality of Astúrias Health Service, Spain: Miguel Prieto; (i) Fundació Lliga per a La Investigació i Prevenció Del Cáncer, Tarragona, Spain: Francina Saladié, Jaume Galceran; (j) Hospital Clinic, Barcelona, Spain; Xavier Bargalló, Isabel Torá-Rocamora; (k) Vallés Oriental Breast Cancer Early Detection Program, Spain; Lupe Peñalva; (l) Catalonian Cancer Strategy, Barcelona, Spain: Josep Alfons Espinàs.

The authors also acknowledge the dedication and support of the Individualized Risk (IRIS) Study Group leaded by Marta Román (mroman@parcdesalutmar.cat) and listed here in alphabetical order and grouped by institution: (a) IMIM (Hospital Del Mar Medical Research Institute), Barcelona, Spain: Rodrigo Alcantara, Xavier Castells, Laia Domingo, Javier Louro, Margarita Posso, Maria Sala, Ignasi Tusquets, Ivonne Vazquez, Mar Vernet-Tomas; (b) Corporació Sanitària Parc Taulí, Sabadell, Spain: Marisa Baré, Javier del Riego; (c) Catalan Institute of Oncology, Barcelona, Spain: Llucia Benito, Carmen Vidal (d) Hospital Santa Caterina, Girona, Spain: Joana Ferrer; (e) Catalan Institute of Oncology, Girona, Spain: Rafael Marcos-Gragera; (f) Hospital de la Santa Creu i Sant Pau, Barcelona, Spain: Judit Solà-Roca, María Jesús Quintana; (g) General Directorate of Public Health, Government of Cantabria, Spain: Mar Sánchez; (h) Principality of Astúrias Health Service, Spain: Miguel Prieto; (i) Fundació Lliga per a La Investigació i Prevenció Del Cáncer, Tarragona, Spain: Francina Saladié, Jaume Galceran; (j) Hospital Clinic, Barcelona, Spain; Xavier Bargalló, Isabel Torá-Rocamora; (k) Vallés Oriental Breast Cancer Early Detection Program, Spain; Lupe Peñalva; (l) Catalonian Cancer Strategy, Barcelona, Spain: Josep Alfons Espinàs.

The authors also acknowledge the help of the (l) Biomedical Informatics Research Unit (GRIB) of the UPF; Alfons Gonzalez-Pauner, Ferran Sanz and (m) the Cardiovascular epidemiology and genetics group of the IMIM; Jaume Marrugat, Isaac Subirana, Joan Vila.

Javier Louro is a Ph.D. candidate at the Methodology of Biomedical Research and Public Health program, Universitat Autònoma de Barcelona (UAB), Barcelona, Spain

## Author Contributions

**Conceptualization:** Javier Louro, Marta Román, Margarita Posso, Xavier Castells.

**Data curation:** Javier Louro, Marta Román, Ivonne Vázquez, Ana Rodriguez-Arana.

**Formal analysis:** Javier Louro.

**Funding acquisition:** Marta Román, Xavier Castells.

**Investigation:** Javier Louro, Marta Román, Margarita Posso, Laia Domingo, Rafael Marcos-Gragera, María Vernet-Tomas, Xavier Castells.

**Methodology:** Javier Louro, Marta Román, Margarita Posso, Laia Domingo, Xavier Castells.

**Project administration:** Javier Louro, Xavier Castells.

**Resources:** Javier Louro, Francina Saladié, M. Jesús Quintana, Marisa Baré, Rafael Marcos-Gragera.

**Supervision:** Marta Román, María Vernet-Tomas, Maria Sala.

**Validation:** Maria Sala.

**Writing – original draft:** Javier Louro.

**Writing – review & editing:** Javier Louro, Marta Román, Xavier Castells.

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
