## [Decision Letter · Decision Letter 0]

11 Nov 2020

PONE-D-20-29968

Developing and Validating an Individualized Breast Cancer Risk Prediction Model for Women Attending Breast Cancer Screening

PLOS ONE

Dear Dr. Román,

Thank you for submitting your manuscript to PLOS ONE. After careful consideration, we feel that it has merit but does not fully meet PLOS ONE’s publication criteria as it currently stands. Therefore, we invite you to submit a revised version of the manuscript that addresses the points raised during the review process.

Please submit a point by point response to each of the reviewers' comments. Pay special attention to the comments regarding clarification on the motivation for this paper, including how this will add to the literature and move the field of breast cancer risk prediction forward. In addition, there were numerous comments about the statistical methods that need more detail and clarification in the manuscript.

We look forward to receiving your revised manuscript.

Kind regards,

Erin J A Bowles

Academic Editor

PLOS ONE

Journal Requirements:

2.We note that you have indicated that data from this study are available upon request. PLOS only allows data to be available upon request if there are legal or ethical restrictions on sharing data publicly. For information on unacceptable data access restrictions, please see http://journals.plos.org/plosone/s/data-availability#loc-unacceptable-data-access-restrictions.

3. One of the noted authors is a group or consortium [BELE and IRIS Study Groups]. In addition to naming the author group, please list the individual authors and affiliations within this group in the acknowledgments section of your manuscript. Please also indicate clearly a lead author for this group along with a contact email address.

5. We note you have included a table to which you do not refer in the text of your manuscript. Please ensure that you refer to Table 4 in your text; if accepted, production will need this reference to link the reader to the Table.

Reviewers' comments:

Reviewer's Responses to Questions

**Comments to the Author**

1. Is the manuscript technically sound, and do the data support the conclusions?

Reviewer #1: Yes

Reviewer #2: Partly

2. Has the statistical analysis been performed appropriately and rigorously? 

Reviewer #1: Yes

Reviewer #2: I Don't Know

3. Have the authors made all data underlying the findings in their manuscript fully available?

Reviewer #1: Yes

Reviewer #2: No

4. Is the manuscript presented in an intelligible fashion and written in standard English?

Reviewer #1: Yes

Reviewer #2: Yes

5. Review Comments to the Author

Reviewer #1: The manuscript report the creation of a new breast cancer risk prediction model developed among a large mammography cohort in Spain. The aim of the study, rationale and potential implications of their new risk model are poorly outlined, making it difficult to evaluate the paper's impact. This also makes it difficult to evaluate whether the methods are appropriate to answer their question.

I found the introduction failed to clearly communicate the motivation for the study. The authors first say that existing breast cancer risk prediction models did not specifically focus on women eligible for mammography screening. (Line 83) However, the Gail model and BCSC models were both developed among women undergoing mammography screening. Granted, these studies were from the US where women begin screening at a younger age, but they account for age in the risk estimates. So I think this motivation for a new risk model is weak.

Next the introduction talks about the fact that only 1 model was focused on personalizing screening, and that it only provided short term risk estimates and does not account for time varying covariates. So the authors say there is a need for short term and long term risk estimates from large cohorts. However, short term vs. long term is not defined, nor the time varying covariates that may be important to consider. Also, if we are interested in personalized screening, does short term or long term risk matter more? Which should we use for changing screening behavior? There is no discussion of this important point.

Then in the final paragraph of the introduction the authors state that the aim is to estimate biennial risk of breast cancer. Which seems like the focus is on short term risk.

A better motivation for this risk model is to include detailed information on BBD type and mammographic findings, which the existing risk models tend not to use. I do think that would be valuable.

Reading between the lines, it seems like you do not have sufficient information to run the other risk models (reproductive risk factors, breast density) so you wanted to make a model using the variables that you do have. I think this is fine, but I would clearly communicate that. One reason the existing risk models have not been well implemented for personalized screening is because it is difficult to collect all of the necessary risk factors in practice, so I think a simpler model like this could be useful, particularly if it performs as well as other models that include many variables that may not be available. This point is not at all addressed in the paper.

When we get to the methods, the model is a Cox model with time varying covariates. Risk predictions for 2 year intervals out to 20 years are validated. In the results the 10 year risk prediction is reported first, which was surprising as I was expected the 2 year risk estimate to be the focus from the intro. The performance is evaluated for 2 year intervals, and AUC is moderate and calibration is good. However, it is unclear how you would apply this 2 year model in practice. The 2 year performance is pretty good. Do you evaluate a woman's 2 year risk, and then alter the screening interval accordingly? Do you look at the 10 or 20 year risk? Its unclear why having risk estimates for every 2 year interval is more useful than selecting one interval.

The discussion does not elaborate on the clinical impact of this model. It also does not compare the model performance to performance estimates for existing models in the literature. And the model performance was not directly compared to model performance of existing models, which would be very helpful. It is also concerning that breast density was not available, as it would be very important to determine whether the mammography findings provided additional predictive value beyond density, or if they were more predictive than density. This is an important point that was unable to be addressed in the study. In the end the paper provided another breast cancer risk prediction model that appears about the same as other models in terms of discriminatory accuracy, which is moderate. This may be useful for some institutions where other risk factors are not available, but doesn't really move the needle in terms of improving our ability to identify high risk women who need more intensive screening.

Reviewer #2: General

This paper develops a breast cancer risk model for women aged 50-69y, using data from a cohort of 121,969 women attending 2y mammography screening 1995-2015 at a centre in Barcelona, Spain. 60% of the sample are used to develop a model based on age, family history, benign breast disease and mammographic features associated with abnormalities reported by interpreting radiologists. Performance is assessed in a hold-out (40%) sample using calibration coefficients and AUC. The authors conclude they have developed a model for short- and long-term risk assessment, that could be used to guide screening strategies. It appears to be a valuable cohort and data set. The most interesting aspect of the analysis to me was use of the radiological abnormalities for risk assessment.

Major

1. It is not clear that the model is suitable for long-term predictions. You only evaluated it when the variables were updated through time (every two years)? For example, I doubt that the mammographic abnormalities used are long-term predictors? I would expect that they indicate that some cancers were missed at the screen. No data are presented to assess this?

2. The abstract reports that the model is "validated" and "could help to guiding individualized screening strategies". This seems too strong. For example, you have not applied the model to a different setting than in which it was developed, nor tried it out in a different time epoch. Further, some of the results are at odds with the literature, including the risk associated with proliferative benign disease, which seems to high. Such aspects would make be wary about proposing it as anything more than a working model to be tested / improved.

3. The statistical methods are inefficient. Rather than split the data, one could have considered cross validation or bootstrapping to estimate optimism in the estimates, for example. There is very little apparent model / variable selection done in this work. Why did you split the data in this way?

4. Some of the commentary on previous work does not seem right, specific comments below.

5. I didn't follow all the methods used, some clarification would be helpful. Specific comments below.

6. It is a shame that the data cannot be made available due to confidentiality. There are precedents for researchers releasing such data used to fit risk models. For example, you can access a modified version of the BCSC data used for their model, where categories have been coded (e.g. not individual year of age). I'd encourage the authors to consider trying to do this if at all possible. What are the confidentiality issues here? It would also be worth making your code available, for transparency of statistical methods used.

Minor

1. Introduction "To date, only one model has specifically aimed to predict a women's individual risk..", there are probably hundreds, and you have referenced more than one. I don't follow what you mean by this.

2. Methods para 1. National screening started 2006. Confirm that organised program started in Barcelona 1995? What was coverage through time in your cohort?

3. Methods para 2. Centres routinely gather info on family history .. How do they do this? e.g. self report for family history? This stayed the same 1995-2015?

4. Very little missing data leading to exclusions, but do you know reasons?

5. Why did you define family history in the way chosen? Before or after looking at data?

6. Putting atypia with usual type will bias your risk estimate, and make it less useable in practice. Seems a bad thing to do for utility of the final model and needs more acknowledgement (and ideally do something to rectify).

7. Do you know reasons for unknown biopsy path result? Related to epoch?

8. Invasive / DCIS. WOuld be useful to assess heterogeneity of results by this? Would be particularly interesting regarding calcs as risk factor. At the very least I think it would be helpful to provide information on the number invasive / DCIS by age and calendar time entry?

9. Did you consider other risk factors for your model, or only those reported? If others, which ones.

10. What is a partly conditional Cox model? Is it just a Cox model?

11. How did you incorporate changing risk factors through time? As a time-dependent covariate?

12. What robust confidence intervals (method).

13. Explain more your "at risk" definition. I don't follow "2 years after the last mammographic examination for follow-up of interval cancer cases".

14. What were reasons for censoring? How many for each reason. e.g. Did anyone die? What if a woman did not attend her screening visit? What if she was older than 69y?

15. How did you model age? In piecewise constant 5y intervals? Why?

16. Did you AUC consider follow-up time? e.g. Some will have entered cohort later than others. It appears that you look at 2y risk as predictor and yes/ no cancer in that period. Multiple values for each person. Did you adjust for loss of independence due to this? Standard Hanley and McNeil would not?

17. Please report actual p values, not p<0.05 etc.

18. Please report confidence intervals on calibration coefficients.

19. In text it appears a lot of women had biopsies with unknown diagnosis (almost one quarter?). Why so many? When was this? At entry or at any time throughout followup?

20. The distribution of 10y risk show none with >8% 10y risk. This is a cutoff used by clinical guidelines in UK to identify women at high risk. Why none? Is the model useful for intended purpose if no women at high risk are identified?

21 Discussion 238. ".case-control design... may overestimate.." why?

22. Discussion BBD. Several models include this, not only one. For example, the IBIS model you reference, BCRAT includes information on biopsies, there are others.

23. Table 1, p-value < 0.05 for all - a bit meaningless. Suggest either drop completely, or put the actual p-value in the table.

24. Almost 30% women had a mammographic abnormality. Is this consistent with what you'd expect? Can you put this into context? Does this mean BI-RADS category 2+? Did you look at risk based BIRADS 3+ (i.e. recalled or not)?

25. Table 4. I don't think you give sufficient detail for me to know how you calculated this table (methods). In particular, how did you estimate expected risk? Is it based on updating risk factors through time? Please provide enough detail in the methods for reproducibility.

26. Finally, worth verifying you have included everything in the TRIPOD checklist.

6. PLOS authors have the option to publish the peer review history of their article (what does this mean?). If published, this will include your full peer review and any attached files.

Reviewer #1: No

Reviewer #2: No

---

## [Author Response · Author response to Decision Letter 0]

4 Jan 2021

Reviewers' comments:

Review Comments to the Author

Reviewer #1: 

The manuscript report the creation of a new breast cancer risk prediction model developed among a large mammography cohort in Spain. The aim of the study, rationale and potential implications of their new risk model are poorly outlined, making it difficult to evaluate the paper's impact. This also makes it difficult to evaluate whether the methods are appropriate to answer their question.

1. I found the introduction failed to clearly communicate the motivation for the study. The authors first say that existing breast cancer risk prediction models did not specifically focus on women eligible for mammography screening. (Line 83) However, the Gail model and BCSC models were both developed among women undergoing mammography screening. Granted, these studies were from the US where women begin screening at a younger age, but they account for age in the risk estimates. So I think this motivation for a new risk model is weak.

Response: 

We thank the reviewer for this comment. We agree that it was not clear what we wanted to highlight, which was that very few studies have been created with the objective of estimating the risk of women targeted to breast cancer screening in order to offer them personalized strategies. We have added a new sentence to clarify it.

New text in introduction (underlined): However, most of these models have not been specifically developed to estimate the risk of women targeted for breast cancer screening in order to offer them personalized strategies.

2. Next the introduction talks about the fact that only 1 model was focused on personalizing screening, and that it only provided short term risk estimates and does not account for time varying covariates. So the authors say there is a need for short term and long term risk estimates from large cohorts. However, short term vs. long term is not defined, nor the time varying covariates that may be important to consider. Also, if we are interested in personalized screening, does short term or long term risk matter more? Which should we use for changing screening behavior? There is no discussion of this important point. Then in the final paragraph of the introduction the authors state that the aim is to estimate biennial risk of breast cancer. Which seems like the focus is on short term risk.

Response: 

We thank the reviewer for this suggestion. What we understand by biennial risk, is not only the risk at 2 years, but every 2 years (at 2, at 4, at 6, at 8, etc ...). It is interesting to observe the different periods of time, and it is necessary to have estimates longer than 2 years if it is intended that some women are screened with a periodicity higher than two years (for example screening the low-risk group every three or four years as some articles propose). We have added a new text to clarify it.

New text in introduction (underlined): This model only estimates the two-year risk, which could lead to bias as one of the aims proposed in breast cancer screening personalization is to see which women are at a lower risk in order to extend their screening period to three or four years. Therefore, if new breast cancer risk models are developed with the aim of analyzing the possibilities offered by personalized screening strategies, it would be interesting to estimate the biennial risk of each woman, in other words, to obtain estimators not only at 2 years, but also every two years (2, 4, 6, 8... up to 20 years, which is the total time a woman is screened). This will help to better understand the different possibilities of screening strategies and will allow to observe the differences in the validation of the model estimators for the different time horizons.

3. A better motivation for this risk model is to include detailed information on BBD type and mammographic findings, which the existing risk models tend not to use. I do think that would be valuable.

Response: 

We thank the reviewer for this contribution. We have modified the text in the introduction to emphasize that both benign breast disease type and mammographic findings are important variables when creating models for this purpose.

New text in introduction (underlined): These new risk models should include a limited and feasible number of variables for the proposed objective, for example, detailed information on the type of previous benign breast disease or previous mammographic characteristics, which existing risk models tend not to use.

4. Reading between the lines, it seems like you do not have sufficient information to run the other risk models (reproductive risk factors, breast density) so you wanted to make a model using the variables that you do have. I think this is fine, but I would clearly communicate that. One reason the existing risk models have not been well implemented for personalized screening is because it is difficult to collect all of the necessary risk factors in practice, so I think a simpler model like this could be useful, particularly if it performs as well as other models that include many variables that may not be available. This point is not at all addressed in the paper.

Response: 

We thank the reviewer for such an interesting contribution. We have added a sentence explaining this fact in the discussion.

New text in discussion (underlined): The model showed modest discrimination with a maximum AUC of 64.7%. Discriminatory accuracy in breast cancer risk prediction models is usually low because a substantial proportion of cases are diagnosed in women with no known risk factors and the AUC of the different models vary between 60 and 70% [14]. This is clearly in contrast with prediction models for other diseases, such as cardiovascular disease, which achieve good discrimination [35, 36]. However, the model presented in this paper performed as well as other models that include many other risk factors that were not available in this study. As one of the reasons why the existing risk models have not been implemented for personalized screening is that it is difficult to collect all of the necessary risk factors in practice, a simpler model like the one we present could be useful.

5. When we get to the methods, the model is a Cox model with time varying covariates. Risk predictions for 2 year intervals out to 20 years are validated. In the results the 10 year risk prediction is reported first, which was surprising as I was expected the 2 year risk estimate to be the focus from the intro. The performance is evaluated for 2 year intervals, and AUC is moderate and calibration is good. However, it is unclear how you would apply this 2 year model in practice. The 2 year performance is pretty good. Do you evaluate a woman's 2 year risk, and then alter the screening interval accordingly? Do you look at the 10 or 20 year risk? Its unclear why having risk estimates for every 2 year interval is more useful than selecting one interval.

Response: 

We thank the reviewer for this comment. 

First, we have changed the order of tables 3 and 4. We realized that it was confusing, that first the specific validation of year 10 was shown and the general validation of all the other time horizons was shown after. 

In previous table 3 (now table 4), we calculated the E/O ratio estimates in risk factor subgroups for each one of the time horizons, but since we wanted to show only one, to have a more easily interpretable table, we chose the 10-year estimate as a summary. 

In the first time horizons (2-, 4- 6-) the number of observed events was lower, being even 0 for some subcategories, which gave error in the estimation of the E/O ratio or wide confidence intervals. When we look at the last ones (20-, 18-, 16-...) we have a lower sample size, because not all of the population has been observed during the 20 years of follow-up (the average of follow-up is slightly more than 7.5 years). 

Therefore, we finally decide to show the 10-year estimate, which has a balanced number of both participants and events observed for each subcategory and which would be perfectly valid for the objective of the study. 

We have added a new paragraph in the discussion to clarify this.

New text in discussion (underlined): In table 4 we saw the calibration of the 10-year estimates from the model in risk factor subgroups. We also performed the E/O ratio estimates in risk factor subgroups for each one of the time horizons proposed. We only showed the 10-year estimates since showing all of them could be confusing. We showed the 10-year estimates since they have a good balance between the number of events observed (in the first time horizons some subcategories have a low number of observed events was lower) and the number of people observed (in the last time horizons we have some lost to follow–up, as the mean time of follow-up is 7.5 years).

6. The discussion does not elaborate on the clinical impact of this model. It also does not compare the model performance to performance estimates for existing models in the literature. And the model performance was not directly compared to model performance of existing models, which would be very helpful. 

Response: 

We thank the reviewer for this suggestion. As we specified in comment 4, we have clarified in the discussion that the model performed as well as many other models that have many more variables and so are harder to implement.

New text in discussion (underlined): The model showed modest discrimination with a maximum AUC of 64.7%. Discriminatory accuracy in breast cancer risk prediction models is usually low because a substantial proportion of cases are diagnosed in women with no known risk factors and the AUC of the different models vary between 60 and 70% [14]. This is clearly in contrast with prediction models for other diseases, such as cardiovascular disease, which achieve good discrimination [35, 36]. However, our model performed as well as other models that include many other risk factors that were not available in this model. As one of the reasons why the existing risk models have not been well implemented for personalized screening is that it is difficult to collect all of the necessary risk factors in practice, a simpler model like the one we present could be useful. 

7. It is also concerning that breast density was not available, as it would be very important to determine whether the mammography findings provided additional predictive value beyond density, or if they were more predictive than density. This is an important point that was unable to be addressed in the study. In the end the paper provided another breast cancer risk prediction model that appears about the same as other models in terms of discriminatory accuracy, which is moderate. This may be useful for some institutions where other risk factors are not available, but doesn't really move the needle in terms of improving our ability to identify high risk women who need more intensive screening.

Response: 

We thank the reviewer for this comment. Regarding breast density, we agree that is a huge limitation and we tried to reflect it in the discussion: “First, a major weakness is the lack of information on breast density, which was not systemically collected as part of screening data in the participating centers. Previous models estimating individual breast cancer risk have shown that the addition of breast density improved the discriminatory power of the models [9, 17, 38, 39]. Dense breasts confer women with a higher risk of breast cancer but are also associated with a higher risk of false-positive results, masking, and interval cancers [40]”

Eriksson et al (See Eriksson M, Czene K, Pawitan Y, Leifland K, Darabi H, Hall P. A clinical model for identifying the short-term risk of breast cancer. Breast Cancer Res. 2017;19(1):29.) proved that mammography findings provide additional predictive value beyond density, incrementing the AUC of the model with density in 6 points after adding calcifications and masses. 

Future work will focus in gathering mammographic density to improve our model. Even so, we have added a sentence specifying that this lack of information in the model could be interesting for institutions that cannot obtain density or SNPS.

New text in discussion (underlined): This lack of both variables may be useful for some institutions where these risk factors are not available.

 

Reviewer #2: General

This paper develops a breast cancer risk model for women aged 50-69y, using data from a cohort of 121,969 women attending 2y mammography screening 1995-2015 at a centre in Barcelona, Spain. 60% of the sample are used to develop a model based on age, family history, benign breast disease and mammographic features associated with abnormalities reported by interpreting radiologists. Performance is assessed in a hold-out (40%) sample using calibration coefficients and AUC. The authors conclude they have developed a model for short- and long-term risk assessment, that could be used to guide screening strategies. It appears to be a valuable cohort and data set. The most interesting aspect of the analysis to me was use of the radiological abnormalities for risk assessment.

Major

1. It is not clear that the model is suitable for long-term predictions. You only evaluated it when the variables were updated through time (every two years)? For example, I doubt that the mammographic abnormalities used are long-term predictors? I would expect that they indicate that some cancers were missed at the screen. No data are presented to assess this?

Response: 

We thank the reviewer for this interesting comment. In previous papers, (See Castells X, Tora I, Posso M et al. Risk of Breast Cancer in Women with False-Positive Results according to Mammographic Features. Radiology 2016, Vol 280, No.2), we have seen that breast cancer risk after mammographic abnormalities is maintained for at least 15 years. Therefore, we think that these abnormalities might be used for medium and long-term predictions. In addition, in our cohort only 59% of cancers after a lesion appear in the same breast (ipsilateral), and in the case of other recognized papers such as the Hartmann study of benign breast disease (Hartmann L, Sellers T, Frost MH et al. Benign Breast Disease and the Risk of Breast Cancer, N Engl J Med 2005; 353:229-237), this proportion is even lower (55%). Because of this evidence we think that the reason might not only be that cancers were missed at the screen but that these abnormalities act as biomarkers and can really be used to estimate the future risk.

2. The abstract reports that the model is "validated" and "could help to guiding individualized screening strategies". This seems too strong. For example, you have not applied the model to a different setting than in which it was developed, nor tried it out in a different time epoch. Further, some of the results are at odds with the literature, including the risk associated with proliferative benign disease, which seems to high. Such aspects would make be wary about proposing it as anything more than a working model to be tested / improved.

Response: 

We thank the reviewer for this suggestion. We have changed “validated” by “internally validated” all over the text, including in abstract and conclusions, to make clear that is not validated with a different setting. Please refer to the updated version of the manuscript for text changes. In the discussion with the sentence “However, external validation of the results is needed to verify the predictive performance of our risk model” we want to clarify that one of the future objectives is validate it with another setting.

3. The statistical methods are inefficient. Rather than split the data, one could have considered cross validation or bootstrapping to estimate optimism in the estimates, for example. There is very little apparent model / variable selection done in this work. Why did you split the data in this way?

Response: 

We thank the reviewer for this comment. We believe that splitting the data in a totally randomized way might be a correct way to make an internal validation. We agree that there are other correct and interesting forms to do it, as cross validation and bootstrapping. Other well-known and widely used models such as Breast Cancer Surveillance Consortium (BCSC) (see Tice JA, Miglioretti DL, Li CS et al. Breast Density and Benign Breast Disease: Risk Assessment to Identify Women at High Risk of Breast Cancer. J Clin Oncol. 2015;33(28):3137-43) used split validation to perform the internal validations of their models. We agree that we should clarify the other options in discussion, and we have added a brief sentence and new references to do it. 

New text in discussion (underlined): To perform internal validation we split our cohort in two sets, the model creation subcohort, to perform the analysis and development of the model and the validation subcohort, to perform the internal validation of the model. This technique known as split validation is common for this type of models [9] but cross validation or bootstrapping could also have been performed [35, 36].

35. Steyerberg EW, Harrell FE, Jr, Borsboom GJ, Eijkemans MJ, Vergouwe Y, Habbema JD. Internal validation of predictive models: efficiency of some procedures for logistic regression analysis. J Clin Epidemiol. 2001;54(8):774–781.

36. Efron B, Tibshirani R. Improvements on cross-validation: The .632+ bootstrap method. J Amer Statist Assoc. 1997;92(438):548–560.

4. Some of the commentary on previous work does not seen m right, specific comments below.

Response: 

We thank the reviewer for these contributions. Specific comments are answered below.

5. I didn't follow all the methods used, some clarification would be helpful. Specific comments below.

Response: 

We thank the reviewer for these contributions. Specific comments are answered below.

6. It is a shame that the data cannot be made available due to confidentiality. There are precedents for researchers releasing such data used to fit risk models. For example, you can access a modified version of the BCSC data used for their model, where categories have been coded (e.g. not individual year of age). I'd encourage the authors to consider trying to do this if at all possible. What are the confidentiality issues here? It would also be worth making your code available, for transparency of statistical methods used.

Response: 

We thank the reviewer for this contribution. We have uploaded the database to the Harvard Dataverse online repository. 

The data is accessible with DOI: https://doi.org/10.7910/DVN/3T7HCH

Minor

1. Introduction "To date, only one model has specifically aimed to predict a women's individual risk...”, there are probably hundreds, and you have referenced more than one. I don't follow what you mean by this.

Response: 

We thank the reviewer for this question. We realize that it is not clear what we wanted to highlight, which is that very few studies have been created with the objective of estimating the risk of women targeted to breast cancer screening to offer them personalized strategies. 

We have added new text to clarify it. Please see response to question #1 of reviewer 1.

New text in introduction (underlined): However, most of these models have not been specifically developed to estimate the risk of women targeted for breast cancer screening in order to offer them personalized strategies.

2. Methods para 1. National screening started 2006. Confirm that organised program started in Barcelona 1995? What was coverage through time in your cohort?

Response: 

Breast cancer screening in Spain started in 1990 in a single setting and expanded until it became nationwide in 2006. Breast Cancer Screening in the city of Barcelona started in 1992 in one area. In those centers from which information has been used for this study, breast cancer screening started in 1995. In Catalonia, we reached 100% coverage in 2004. You can find this and more information about breast cancer screening in Spanish in reference 19 of the article: 

Ascunce N, Salas D, Zubizarreta R, Almazan R, Ibanez J, Ederra M, et al. Cancer screening in Spain. Ann Oncol. 2010;21 Suppl 3:iii43-51.

3. Methods para 2. Centres routinely gather info on family history. How do they do this? e.g. self report for family history? This stayed the same 1995-2015?

Response: 

We thank the reviewer for this comment. Family history information is obtained through a face-to-face interview conducted by trained professionals at the time of mammography screening. These questionnaires have been reported systematically since the implementation of the programs, that is, during the 20 years study period. We have clarified this information in the manuscript. Please refer to updated text.

New text in Material and Methods (underlined): Information on family history and history of prior breast biopsies was self-reported and collected from face-to-face interviews conducted by trained professionals at the time of mammography screening. This information was consistently collected over the 20 years study period.

4. Very little missing data leading to exclusions, but do you know reasons?

Response: 

We agree with the reviewer that the number of missing data leading to exclusions is quite small. The questionnaire is systematically gathered by a professional at the time of the first mammography and is mandatory before women get done their mammograms. It is for this reason that during all the years of follow-up, very few questionnaire variables have been left as missing by the professional.

5. Why did you define family history in the way chosen? Before or after looking at data?

Response:

We thank the reviewer for this comment. The definition of family history as a first degree relative is the initial definition of the questionnaire systematically filled in by a professional since 1995. As this is a retrospective cohort study, we assessed the family history information as gathered on the questionnaires. No manipulation of data was done by the researchers in this study.

6. Putting atypia with usual type will bias your risk estimate, and make it less useable in practice. Seems a bad thing to do for utility of the final model and needs more acknowledgement (and ideally do something to rectify).

Response: 

We thank the reviewer for this observation. We absolutely agree, and we are aware that combining both proliferative lesions with and without atypia into a single category might be a limitation of the study. We have tried to reflect this issue in the discussion. Because of small number of subsequent breast cancer cases among those with a proliferative lesion with atypia, we merge both into a single category. We have added a paragraph in the limitations part of the discussion section specifically addressing this issue.

New text in discussion (underlined): Also, as a consequence of the small number of subsequent breast cancer cases among those women with a proliferative BBD with atypia, we merged proliferative BBD with and without atypia into a single category which might make the model less usable in practice.

7. Do you know reasons for unknown biopsy path result? Related to epoch? 

Response: 

We thank the reviewer for this comment. As women reported BBD before the start of screening, but with no pathology results available, we created the category “Prior biopsy, unknown diagnosis” to do not lose this information. Other studies like the BCSC (see Tice JA, Miglioretti DL, Li CS et al. Breast Density and Benign Breast Disease: Risk Assessment to Identify Women at High Risk of Breast Cancer. J Clin Oncol. 2015;33(28):3137-43) have used this way of categorizing. We have clarified this text in the manuscript.

New text in material and methods (underlined): If women reported having had a biopsy before the start of the screening but no pathology results were available, the biopsy was classified as having a prior biopsy, unknown diagnosis.

8. Invasive / DCIS. Would be useful to assess heterogeneity of results by this? Would be particularly interesting regarding calcs as risk factor. At the very least I think it would be helpful to provide information on the number invasive / DCIS by age and calendar time entry?

Response: 

We thank the reviewer for this contribution. We agree that this subanalisis could be particularly interesting; unfortunately, information on the DCIS/invasive status of the tumours was not initially collected when information for this study was gathered. As we work with an anonymized data base it seems complex to add this information at this stage.

9. Did you consider other risk factors for your model, or only those reported? If others, which ones.

Response: 

We absolutely agree with the reviewer, it would have been desirable to have information on other risk factors, however, as we use a retrospective cohort data, the available information on other risk factors is limited. We have tried to make clear in the discussion that this is one of the major limitations of the model. Future work will be focused on obtain the breast density and update the model. 

Nevertheless, our internal validation showed that our model performed similar than the previous risk models and include a fewer number of risk factors. One of the reasons why breast cancer screening is not using the existing risk models to implement personalized screening is that it is difficult to collect all the necessary risk factors in practice. We think that for that reason a simpler model like the one we present could be useful. We have added a next text in the discussion to include this information.

New text in discussion (underlined): However, the model presented in this paper performed as well as other models that include many other risk factors that were not available in this study. As one of the reasons why the existing risk models have not been implemented for personalized screening is that it is difficult to collect all of the necessary risk factors in practice, a simpler model like the one we present could be useful.

10. What is a partly conditional Cox model? Is it just a Cox model?

Response: 

A partly conditional cox model is an extension of the classical proportional hazards cox model that uses repeated measures and allows to incorporate changes in these risk factors over time (for example, if a woman has not had a benign breast disease in the first 10 years of follow-up, but has had one for 10 years thereafter, you can take this information into account). 

You can consult more information about this survival model in the article:

 Zheng YZ, Heagerty PJ. Partly conditional survival models for longitudinal data. Biometrics. 2005;61:379–391. Maziarz M, Heagerty P, Cai T, Zheng Y. 

In addition, in this article: 

On longitudinal prediction with time-to-event outcome: Comparison of modeling options. Biometrics. 2017 Mar;73(1):83-93. doi: 10.1111/biom.12562. Epub 2016 Jul 20. PMID: 27438160; PMCID: PMC5250577.

You also can find a comparison between these models, the joint models and the partly conditional generalized linear models.

We have modified the manuscript to clarify this issue:

New text in Material and Methods (underlined): We used partly conditional Cox proportional hazards regression, an extension of the standard Cox model, to incorporate changes in these risk factors over time. Robust standard errors were used to estimate 95% confidence intervals.

11. How did you incorporate changing risk factors through time? As a time-dependent covariate?

Response: 

Information on risk factors through time were incorporated as time-dependent covariates. The partly conditional cox model mentioned allowed us to perform the analysis by screening participation instead of by women. This model takes into account the correlated observations between the different screening participations of the same woman during all their screening lifespan.

The model was performed the statistical software R version 3.4.3 (Development Core Team, 2014) with the package partlyconditional. 

More information of this package can be found at:

https://github.com/mdbrown/partlyconditional

12. What robust confidence intervals (method).

Response: 

In particular, we used the robust standard error reported by the Huber sandwich estimator to create the robust confidence intervals. This is a standard estimation method to obtain robust estimates and is the one reported both by the Standard Cox function (coxph function of the R package survival) and by the model used for the Partly Conditional Cox model (partlyconditional package).

More information of this package can be found at:

https://github.com/mdbrown/partlyconditional

13. Explain more your "at risk" definition. I don't follow "2 years after the last mammographic examination for follow-up of interval cancer cases".

Response: 

We thank the reviewer for this important contribution. We understand now that it is not clear in the manuscript. If a woman has had a diagnosis of cancer, she will contribute women-years at risk from the date of her first mammogram to the diagnosis of cancer. Those women who do not have a diagnosis of cancer will contribute women-years at risk from the first mammogram to the last mammogram plus two years of follow-up in which we know they have not had a cancer, since we can identify those who have had an interval cancer. We have clarified this issue in the methods section.

New text in Material and Methods (underlined): If a woman has had a diagnosis of cancer, she will contribute women-years at risk from the date of her first mammogram to the diagnosis of cancer. Since we can identify all interval cancers, a woman who has not had a diagnosis of cancer at the end of her follow-up will contribute women-years at risk from the first mammogram to the last mammogram plus 2 years of follow-up.

14. What were reasons for censoring? How many for each reason. e.g. Did anyone die? What if a woman did not attend her screening visit? What if she was older than 69y?

Response: 

We thank the reviewer for this comment. As we extracted our data directly from the population-based screening program databases we are assured of having the maximum follow-up of these women. We collected in a comprehensive and systematic way all the participations that these women have made in the breast cancer screening program.

However, we cannot know exactly what the cause of each loss of follow-up is. As we do all the analysis by screening participation and not by women, and taking into account the time between mammograms, there is not any problem at all for the analysis if a woman skips one of the screening tests but she returns to the screening later.

Of the 121,969 women in our cohort, 63,694 had their last mammogram within the last two years of the study, which are 2014 and 2015. These women were censored at the end of the study period. 20,436 women of the remaining had their last mammogram at age 68 or 69, so these women completed their screening process during the study follow-up.

Of the remaining 37,839 we know that the majority are women who have decided not to participate in the 2014-2015 round or who have changed health areas and therefore are not in our study population. Regarding the women who die, the screening program does not have an exhaustive record of this cause and they appear to us as non-participating women. 

15. How did you model age? In piecewise constant 5y intervals? Why?

Response: 

To build the model we used age as a quantitative variable. By using a partly conditional Cox model we performed the analysis by screening participation instead of by women. Therefore, risk factors through time were incorporated as time-dependent covariates, age included. 

However, to present the results in an easy way we showed age in piecewise constant 5-year intervals in tables. We believe this is a commonly used way to present results (e.g. BCSC, see Tice JA, Miglioretti DL, Li CS et al. Breast Density and Benign Breast Disease: Risk Assessment to Identify Women at High Risk of Breast Cancer. J Clin Oncol. 2015;33(28):3137-43) 

16. Did you AUC consider follow-up time? e.g. Some will have entered cohort later than others. It appears that you look at 2y risk as predictor and yes/ no cancer in that period. Multiple values for each person. Did you adjust for loss of independence due to this? Standard Hanley and McNeil would not?

Response: 

We thank the reviewer for this interesting comment. Yes, we do consider follow-up time at every horizon for every individual in the cohort. In table 3 (previous table 4), we estimated the AUC for every 2-year interval, which means, that we estimated the 20-year risk AUC for those individuals followed for 20 years, the 18-year risk AUC for those followed for at least 18 years, the 16-year risk AUC for those followed for at least 16 years, and so on. The AUC for each time horizon was estimated with all the women in the validation cohort followed at least that time, using the predicted risk of the model and whether she has developed a tumor or not at the specific time horizon. 

17. Please report actual p values, not p<0.05 etc.

Response: 

We thank the reviewer for this suggestion. We have added exact p-values to table 1 for all values higher than 0.001. 

Please refer to the updated version of the manuscript for text changes.

18. Please report confidence intervals on calibration coefficients.

Response: 

We thank the reviewer for this suggestion. We have added confidence intervals on calibration coefficients in the text.

New text in results (underlined): Estimates for the 10-year time horizon showed that the model slightly overestimated breast cancer rates in women with masses (E/O ratio, 1.18; 95%CI: 1.02-1.37) and in women aged 55-59 years (E/O ratio, 1.15; 95%CI: (1.03-1.29) (Table 4). The model also underestimated breast cancer rates in women aged 50-54 years (E/O ratio, 0.83; 95%CI: 0.75-0.94). Because of the small number of breast cancer cases, calibration was overestimated among women with proliferative BBD (E/O ratio, 1.85; 95%CI: 1.00-3.40).

19. In text it appears a lot of women had biopsies with unknown diagnosis (almost one quarter?). Why so many? When was this? At entry or at any time throughout followup?

Response: 

Please see response to question #7 of reviewer 2. As women reported BBD before the start of screening, but with no pathology results available, we created the category “Prior biopsy, unknown diagnosis” to do not lose this information. Other studies like the BCSC (see Tice JA, Miglioretti DL, Li CS et al. Breast Density and Benign Breast Disease: Risk Assessment to Identify Women at High Risk of Breast Cancer. J Clin Oncol. 2015;33(28):3137-43) have used this way of categorizing. We have clarified this text in the manuscript.

New text in introduction (underlined): If women reported having had a biopsy before the start of the screening but no pathology results were available, the biopsy was classified as having a prior biopsy, unknown diagnosis.

20. The distribution of 10y risk show none with >8% 10y risk. This is a cutoff used by clinical guidelines in UK to identify women at high risk. Why none? Is the model useful for intended purpose if no women at high risk are identified?

Response: We thank the reviewer for this interesting contribution. Of the 48,815 women who had their risk estimated at 10 years, only 134 had a risk higher than 8%. This is the reason why we have cut the histogram before, as there was a higher proportion in the group between 1.5% and 2%, the reader could not correctly distinguish these group in the histogram.

We have realized that the figure could be confusing for the reader, and we have modified the axis by adding at the end a “Higher” to make it clearer that people with a higher risk are included in the last break.

21 Discussion 238. ".case-control design... may overestimate.." why?

Response: 

We thank the reviewer for this comment. 

We think that in a research question like the one that we are addressing, which is of longitudinal nature, it might be better to use a longitudinal design as a cohort study. A case control design does not take into account the time at risk that passes before an event occurs and this fact might lead to a bias since the output of a model in this type of design is not the risk, which is the output we are looking for. 

We realize that was not clear in the text, and that “overestimate” is not the right word, and we have modified the text to qualify it. 

New text discussion (underlined): In addition, it used a case-control design to establish risk factors, which may bias the estimates of the short-term association with breast cancer risk.

22. Discussion BBD. Several models include this, not only one. For example, the IBIS model you reference, BCRAT includes information on biopsies, there are others.

Response:

We thank the reviewer for this comment. We realize that it was not clear on the text that we referred only to models that included pathological classification of benign breast disease, and not just the presence or absence of a benign breast disease. We have modified the sentence to try to make it clearer to the reader.

New text discussion (underlined): Only one previous risk model included different estimates for the different categories of the Dupont and Page BBD pathological classification [23-25]. 

23. Table 1, p-value < 0.05 for all - a bit meaningless. Suggest either drop completely, or put the actual p-value in the table.

Response: 

Please see question #17 of reviewer 2. We thank the reviewer for this comment. We have added exact p-values to table 1 for all values higher than 0.001. 

Please refer to the updated version of the manuscript for text changes.

24. Almost 30% women had a mammographic abnormality. Is this consistent with what you'd expect? Can you put this into context? Does this mean BI-RADS category 2+? Did you look at risk based BIRADS 3+ (i.e. recalled or not)?

Response: 

We thank the reviewer for this comment. In our study, 28% of women present at least one suspicious mammographic feature in their entire follow-up. As this 28% refers to a cumulative prevalence and not cross-sectional, and we have a long follow-up (average 7.5 years, maximum 20 years), this value effectively fits with what we expected to find. 

However, not all women with a mammographic abnormality do have a false positive or a benign breast disease. Some mammographic abnormalities may include women with a suspicious mammographic reading that were not recalled for further assessment. These mammographic abnormalities embrace BIRADS category 2. In addition, all women with mammographic abnormalities who are recalled for further assessments were classified as BI-RADS category 3+.

25. Table 4. I don't think you give sufficient detail for me to know how you calculated this table (methods). In particular, how did you estimate expected risk? Is it based on updating risk factors through time? Please provide enough detail in the methods for reproducibility.

Response: We thank the reviewer for this important observation. We agree with this comment, both in table 3 and 4, we must explain how we estimate the expected risk. We have tried to clarify it better by adding the following sentence to methods.

New text methods (underlined): The expected breast cancer rate was calculated as the average of the model predicted risk for each woman in a specific subgroup.

26. Finally, worth verifying you have included everything in the TRIPOD checklist.

Response: We have verified that we have included all the points in the TRIPOD Checklist: Prediction Model Development and Validation (Collins GS, Reitsma JB, Altman DG, Moons KG. Transparent reporting of a multivariable prediction model for individual prognosis or diagnosis (TRIPOD): The TRIPOD statement.)

---

## [Decision Letter · Decision Letter 1]

23 Feb 2021

PONE-D-20-29968R1

Developing and Validating an Individualized Breast Cancer Risk Prediction Model for Women Attending Breast Cancer Screening

PLOS ONE

Dear Dr. Román,

Thank you for submitting your manuscript to PLOS ONE. After careful consideration, we feel that it has merit but does not fully meet PLOS ONE’s publication criteria as it currently stands. Therefore, we invite you to submit a revised version of the manuscript that addresses the points raised during the review process.

Please address all remaining comments from Reviewer #2. Please also do a thorough read of the manuscript and correct all typographical errors (there are several).

We look forward to receiving your revised manuscript.

Kind regards,

Erin J A Bowles

Academic Editor

PLOS ONE

Journal Requirements:

Reviewers' comments:

Reviewer's Responses to Questions

**Comments to the Author**

1. If the authors have adequately addressed your comments raised in a previous round of review and you feel that this manuscript is now acceptable for publication, you may indicate that here to bypass the “Comments to the Author” section, enter your conflict of interest statement in the “Confidential to Editor” section, and submit your "Accept" recommendation.

Reviewer #1: All comments have been addressed

Reviewer #2: (No Response)

2. Is the manuscript technically sound, and do the data support the conclusions?

Reviewer #1: Yes

Reviewer #2: Yes

3. Has the statistical analysis been performed appropriately and rigorously? 

Reviewer #1: Yes

Reviewer #2: Yes

4. Have the authors made all data underlying the findings in their manuscript fully available?

Reviewer #1: Yes

Reviewer #2: Yes

5. Is the manuscript presented in an intelligible fashion and written in standard English?

Reviewer #1: Yes

Reviewer #2: Yes

6. Review Comments to the Author

Reviewer #1: The authors have responded to the reviewer comments in an appropriate way, in particular they have added clarifications to their statements and analytic decisions.

Reviewer #2: Many thanks to the reviewers for addressing my review.

A few small clarification issues from last review below.

--

6. It is a shame that the data cannot be made available due to confidentiality. There are precedents for researchers releasing such data used to fit risk models. For example, you can access a modified version of the BCSC data used for their model, where categories have been coded (e.g. not individual year of age). I'd encourage the authors to consider trying to do this if at all possible. What are the confidentiality issues here? It would also be worth making your code available, for transparency of statistical methods used.

Response:

We thank the reviewer for this contribution. We have uploaded the database to the Harvard Dataverse online repository.

The data is accessible with DOI: https://doi.org/10.7910/DVN/3T7HCH

- Fantastic thank you. Would it also be possible to make available the analysis code used for this paper - for complete reproducibility?

12. What robust confidence intervals (method).

Response:

In particular, we used the robust standard error reported by the Huber sandwich estimator to create the robust confidence intervals. This is a standard estimation method to obtain robust estimates and is the one reported both by the Standard Cox function (coxph function of the R package survival) and by the model used for the Partly Conditional Cox model (partlyconditional package).

More information of this package can be found at:

https://github.com/mdbrown/partlyconditional

- Did you include this in the text?

14. What were reasons for censoring? How many for each reason. e.g. Did anyone die? What if a woman did not attend her screening visit? What if she was older than 69y?

Response:

We thank the reviewer for this comment. As we extracted our data directly from the population-based screening program databases we are assured of having the maximum follow-up of these women. We collected in a comprehensive and systematic way all the participations that these women have made in the breast cancer screening program.

However, we cannot know exactly what the cause of each loss of follow-up is. As we do all the analysis by screening participation and not by women, and taking into account the time between mammograms, there is not any problem at all for the analysis if a woman skips one of the screening tests but she returns to the screening later.

Of the 121,969 women in our cohort, 63,694 had their last mammogram within the last two years of the study, which are 2014 and 2015. These women were censored at the end of the study period. 20,436 women of the remaining had their last mammogram at age 68 or 69, so these women completed their screening process during the study follow-up.

Of the remaining 37,839 we know that the majority are women who have decided not to participate in the 2014-2015 round or who have changed health areas and therefore are not in our study population. Regarding the women who die, the screening program does not have an exhaustive record of this cause and they appear to us as non-participating women.

- Did you include this in the text?

15. How did you model age? In piecewise constant 5y intervals? Why?

Response:

To build the model we used age as a quantitative variable. By using a partly conditional Cox model we performed the analysis by screening participation instead of by women. Therefore, risk factors through time were incorporated as time-dependent covariates, age included.

However, to present the results in an easy way we showed age in piecewise constant 5-year intervals in tables. We believe this is a commonly used way to present results (e.g. BCSC, see Tice JA, Miglioretti DL, Li CS et al. Breast Density and Benign Breast Disease: Risk Assessment to Identify Women at High Risk of Breast Cancer. J Clin Oncol. 2015;33(28):3137-43)

- Did you include in the text? (If you made your analysis code available this would be even more transparent)

16. Did you AUC consider follow-up time? e.g. Some will have entered cohort later than others. It appears that you look at 2y risk as predictor and yes/ no cancer in that period. Multiple values for each person. Did you adjust for loss of independence due to this? Standard Hanley and McNeil would not?

Response:

We thank the reviewer for this interesting comment. Yes, we do consider follow-up time at every horizon for every individual in the cohort. In table 3 (previous table 4), we estimated the AUC for every 2-year interval, which means, that we estimated the 20-year risk AUC for those individuals followed for 20 years, the 18-year risk AUC for those followed for at least 18 years, the 16-year risk AUC for those followed for at least 16 years, and so on. The AUC for each time horizon was estimated with all the women in the validation cohort followed at least that time, using the predicted risk of the model and whether she has developed a tumor or not at the specific time horizon.

- Thank you. But is the risk score a time-varying covariate over the 20y horizon, or you use the baseline assessment? The two are not the same and good to clarify in paper methods.

25. Table 4. I don't think you give sufficient detail for me to know how you calculated this table (methods). In particular, how did you estimate expected risk? Is it based on updating risk factors through time? Please provide enough detail in the methods for reproducibility.

Response: We thank the reviewer for this important observation. We agree with this comment, both in table 3 and 4, we must explain how we estimate the expected risk. We have tried to clarify it better by adding the following sentence to methods.

New text methods (underlined): The expected breast cancer rate was calculated as the average of the model predicted risk for each woman in a specific subgroup.

- Can you say more about this? Predicted risk of breast cancer to what time? There are different ways to do this, cf. https://doi.org/10.1214/19-STS729 . I assume cumulative hazard to time each measurement of predictors / event / censoring, but useful to clarify.

7. PLOS authors have the option to publish the peer review history of their article (what does this mean?). If published, this will include your full peer review and any attached files.

Reviewer #1: No

Reviewer #2: No

---

## [Author Response · Author response to Decision Letter 1]

5 Mar 2021

Comments from the Editors and Reviewers:

6. It is a shame that the data cannot be made available due to confidentiality. There are precedents for researchers releasing such data used to fit risk models. For example, you can access a modified version of the BCSC data used for their model, where categories have been coded (e.g. not individual year of age). I'd encourage the authors to consider trying to do this if at all possible. What are the confidentiality issues here? It would also be worth making your code available, for transparency of statistical methods used.

Previous response:

We thank the reviewer for this contribution. We have uploaded the database to the Harvard Dataverse online repository.

The data is accessible with DOI: https://doi.org/10.7910/DVN/3T7HCH

- Fantastic thank you. Would it also be possible to make available the analysis code used for this paper - for complete reproducibility?

Response: We thank the reviewer for this contribution. Yes, it is possible and we agree it will help for understanding and reproducibility of the paper. We have uploaded the code publicly to github at the following address:

https://github.com/JlouroA/IRISModelCode

We have split the code in two parts, a small part in SPSS where we split the database in the estimation and the validation subcohort and the R code where all the analyses of the paper are done (both model development and validation). 

Note to the reviewer: For a better understanding of the article, we have modified the term "model subcohort" to "estimation subcohort" both in the manuscript and in this response letter to refer to the part of the cohort used for the model development. We believe that speaking of "estimation cohort" and "validation cohort" is much clearer.

12. What robust confidence intervals (method).

Previous response:

In particular, we used the robust standard error reported by the Huber sandwich estimator to create the robust confidence intervals. This is a standard estimation method to obtain robust estimates and is the one reported both by the Standard Cox function (coxph function of the R package survival) and by the model used for the Partly Conditional Cox model (partlyconditional package).

More information of this package can be found at:

https://github.com/mdbrown/partlyconditional

- Did you include this in the text?

Response: Thank you for this suggestion. This was not included in the text and we agree it can improve the clearance of the paper. We have added a new sentence and new reference in the methods section of the article to state the method used to estimate the confidence intervals.

New text in methods (underlined): Robust standard errors were used to estimate 95% confidence intervals using the Huber sandwich estimator [26].

26. Freedman, DA. On the So-Called ‘Huber Sandwich Estimator’ and ‘Robust Standard Errors.’ The American Statistician, vol. 60, no. 4, 2006, pp. 299–302. JSTOR, www.jstor.org/stable/27643806. Accessed 23 Feb. 2021.

14. What were reasons for censoring? How many for each reason. e.g. Did anyone die? What if a woman did not attend her screening visit? What if she was older than 69y?

Previous Response: We thank the reviewer for this comment. As we extracted our data directly from the population-based screening program databases we are assured of having the maximum follow-up of these women. We collected in a comprehensive and systematic way all the participations that these women have made in the breast cancer screening program.

However, we cannot know exactly what the cause of each loss of follow-up is. As we do all the analysis by screening participation and not by women, and taking into account the time between mammograms, there is not any problem at all for the analysis if a woman skips one of the screening tests but she returns to the screening later.

Of the 121,969 women in our cohort, 63,694 had their last mammogram within the last two years of the study, which are 2014 and 2015. These women were censored at the end of the study period. 20,436 women of the remaining had their last mammogram at age 68 or 69, so these women completed their screening process during the study follow-up.

Of the remaining 37,839 we know that the majority are women who have decided not to participate in the 2014-2015 round or who have changed health areas and therefore are not in our study population. Regarding the women who die, the screening program does not have an exhaustive record of this cause and they appear to us as non-participating women.

- Did you include this in the text?

Response: 

We thank the reviewer for this contribution. We agree that this information could be included in the text. We have added a new paragraph in the discussion to make it clearer.

New text in discussion (underlined): Another limitation might be the reason for censoring. Over 52% of women in the cohort had their last mammogram in the last two years of the study follow-up and 17% of women had their last mammogram at ages 68 or 69 years. Most of the remaining 31% are women who did not participate in the 2014-2015 round or who have changed health areas and thus are not in our study population. The screening program does not have an exhaustive record of which women die and, therefore, we cannot differentiate them from non-participating women.

15. How did you model age? In piecewise constant 5y intervals? Why?

Previous response:

To build the model we used age as a quantitative variable. By using a partly conditional Cox model we performed the analysis by screening participation instead of by women. Therefore, risk factors through time were incorporated as time-dependent covariates, age included.

However, to present the results in an easy way we showed age in piecewise constant 5-year intervals in tables. We believe this is a commonly used way to present results (e.g. BCSC, see Tice JA, Miglioretti DL, Li CS et al. Breast Density and Benign Breast Disease: Risk Assessment to Identify Women at High Risk of Breast Cancer. J Clin Oncol. 2015;33(28):3137-43)

- Did you include in the text? (If you made your analysis code available this would be even more transparent)

Response: We have added a new sentence in the text to facilitate the understanding of readers. Thank you for this good suggestion.

New text in methods (underlined): We estimated the age-adjusted hazard ratios (aHR) and the 95% confidence intervals (95%CI) for the breast cancer incidence for each category of family history, previous BBD, and previous mammographic features. Age was included in the model as a continuous variable.

16. Did you AUC consider follow-up time? e.g. Some will have entered cohort later than others. It appears that you look at 2y risk as predictor and yes/ no cancer in that period. Multiple values for each person. Did you adjust for loss of independence due to this? Standard Hanley and McNeil would not?

Previous response:

We thank the reviewer for this interesting comment. Yes, we do consider follow-up time at every horizon for every individual in the cohort. In table 3 (previous table 4), we estimated the AUC for every 2-year interval, which means, that we estimated the 20-year risk AUC for those individuals followed for 20 years, the 18-year risk AUC for those followed for at least 18 years, the 16-year risk AUC for those followed for at least 16 years, and so on. The AUC for each time horizon was estimated with all the women in the validation cohort followed at least that time, using the predicted risk of the model and whether she has developed a tumor or not at the specific time horizon.

- Thank you. But is the risk score a time-varying covariate over the 20y horizon, or you use the baseline assessment? The two are not the same and good to clarify in paper methods.

Response: We thank the reviewer for this comment. This is an important question. We will try to clarify this point:

The AUC is estimated as a baseline assessment, although risk factors are time-changing variables. For example, the 18-year AUC is calculated by taking those women in the validation subcohort followed for at least 18 years and calculating their 18-year risk with their risk factors fixed at their baseline mammogram. 

With the predicted risk and whether they developed a breast cancer or not, we can calculate the AUC of that time horizon (the 18-year risk in this example). 

We apply this calculation for all time horizons independently (2-year, 4-year,…,20-year). 

We agree that this methodological aspect can be misleading. We have added a new sentence in the text to try to make it clearer for the reader.

New text in methods (underlined):

The discriminatory accuracy of our model was assessed by estimating the area under the receiving operating characteristic curve (AUC) for each 2-year interval based on the predicted risks for each woman and each woman’s final outcome whether she developed breast cancer during the time interval or not [29]. The predicted risks were calculated using the model coefficient estimates at the baseline mammogram for those women in the validation cohort who have been followed for a time greater than or equal to the time horizon being estimated. The AUC measured the ability of the model to discriminate between women who will develop breast cancer from those who will not.

25. Table 4. I don't think you give sufficient detail for me to know how you calculated this table (methods). In particular, how did you estimate expected risk? Is it based on updating risk factors through time? Please provide enough detail in the methods for reproducibility.

Previous response: We thank the reviewer for this important observation. We agree with this comment, both in table 3 and 4, we must explain how we estimate the expected risk. We have tried to clarify it better by adding the following sentence to methods.

New text methods (underlined): The expected breast cancer rate was calculated as the average of the model predicted risk for each woman in a specific subgroup.

- Can you say more about this? Predicted risk of breast cancer to what time? There are different ways to do this, cf. I assume cumulative hazard to time each measurement of predictors / event / censoring, but useful to clarify.

Response: We thank the reviewer for this comment. We agree that this methodology is complex and it could be better explained. We will try to clarify:

In table 3 every time horizon uses a different estimate. ; 2-year risk estimate, 4-year risk estimate, etc…

Let’s take the 4-year risk estimate as an example: 

The observed rate was estimated using the Kaplan Meier risk estimate at 4 years of the estimation subcohort. 

The expected rate is calculated as the mean of the predictions of the 4-year risk estimates for all women in the validation subcohort. 

All other time horizons on table 3 are calculated the same way.

However, to compute table 4 we used the 10-year risk estimates as a reference, but the table can be replicated for every time horizon. Using the 10-year estimates is just an illustrative example. 

On table 4, we estimated the expected and observed ratio for each risk factor: With or without family history, with no BBD, with proliferative BBD, etc... 

Similarly to table 3, the observed rate is the Kaplan Meier estimate at 10 years in the specific risk group of the estimation subcohort (Example, only those with family history). The expected breast cancer rate was calculated as the average of the 10-year risk estimates for each woman in the specific risk group of the validation subcohort (Example, only those with family history). 

All code is now available in the aforementioned github repository.

We have added a new sentence in the text to try to make it clearer for the reader. Thanks for the suggestion.

New text in methods (underlined):

To assess calibration, we calculated the ratio between the expected breast cancer rate in the validation subcohort versus the observed rate in the model estimation subcohort. To consider account for censoring, the observed rate was estimated using a the Kaplan-Meier estimator. The expected breast cancer rate was calculated as the average of the model predicted risk for each woman in a subgroup. the risk estimates in the validation subcohort. The expected breast cancer rate in a specific risk group was calculated as the average of the risk estimates for each woman in that risk group of the validation subcohort. The expected-to-observed (E/O) ratio assessed whether the number of women predicted to develop breast cancer from the model matched the actual number of breast cancers diagnosed in the validation subcohort. An E/O ratio of 1.0 indicates perfect calibration.

---

## [Editor Report · Decision Letter 2]

9 Mar 2021

Developing and Validating an Individualized Breast Cancer Risk Prediction Model for Women Attending Breast Cancer Screening

PONE-D-20-29968R2

Dear Dr. Román,

We’re pleased to inform you that your manuscript has been judged scientifically suitable for publication and will be formally accepted for publication once it meets all outstanding technical requirements.

Kind regards,

Erin J A Bowles

Academic Editor

PLOS ONE

---

## [Editor Report · Acceptance letter]

15 Mar 2021

PONE-D-20-29968R2 

Developing and validating an individualized breast cancer risk prediction model for women attending breast cancer screening 

Dear Dr. Román:

I'm pleased to inform you that your manuscript has been deemed suitable for publication in PLOS ONE. Congratulations! Your manuscript is now with our production department. 

Kind regards, 

on behalf of

Dr. Erin J A Bowles 

Academic Editor

PLOS ONE